# FULLDIFFUSION:
# DIFFUSION MODELS WITHOUT TIME TRUNCATION

## ABSTRACT

Diffusion models are predominantly used for generative modeling, which synthesize samples by simulating the reverse process of a stochastic differential equation (SDE) that diffuses data into Gaussian noise. However, when simulating the reverse SDE, the SDE solver suffers from numerical instability near the time boundary; hence, in practice, the simulation is terminated before reaching the boundary point. This heuristic time truncation hinders the rigorous formulation of diffusion models, and requires additional costs of hyperparameter tuning. Moreover, such numerical instability often occurs even in training, especially when using a maximum likelihood loss. Therefore, the current diffusion model heavily relies on the time truncation technique in both training and inference. In this paper, we propose a method that completely eliminates the heuristic of time truncation. Our method eliminates numerical instability during maximum likelihood training by modifying the parameterization of the noise predictor and the noise schedule. We also propose a novel SDE solver that can simulate without time truncation by taking advantage of the semi-linear structure of the reverse SDE. These improvements enable stable training and sampling of diffusion models without relying on time truncation. In our experiments, we tested the effectiveness of our method on the CIFAR-10 and ImageNet-32 datasets by evaluating the test likelihood and the sample quality measured by the Fréchet inception distance (FID). We observe that our method consistently improve performance in both test likelihood and the FID compared to the baseline model of DDPM++.

## 1 INTRODUCTION

Diffusion probabilistic models (Sohl-Dickstein et al., 2015; Ho et al., 2020) and score-based generative models (Song & Ermon, 2019; 2020) have achieved state-of-the-art performance in terms of sample quality especially for image generation. Both models consider to pertub data with a sequence of noise distributions, and generate samples by learning to reverse the diffusion process from noise to data. Song et al. (2020b) have shown that these two types of models can be interpreted using a single framework, which we refer to as *diffusion models* in this paper.

The framework of diffusion models (Song et al., 2020b) involves gradually diffusing the data distribution towards a simple noise distribution, such as the standard Gaussian distribution, using a stochastic differential equation (SDE), and learning the time reversal of this SDE for generative modeling. The reverse-time SDE has an analytic expression which only depends on a time-dependent score function of the perturbed data distribution. This score function can be efficiently estimated by training a neural network (called a score-based model (Song & Ermon, 2019; 2020)) with a weighted combination of score matching losses (Hyvärinen & Dayan, 2005; Vincent, 2011; Song et al., 2020a) as the objective. After training, we can obtain samples from the model by simulating the reverse SDE from a simple noise using the estimated score function.

However, when simulating the reverse-time SDE, the SDE solver suffers from numerical instability near the time boundary. This is mainly because the estimated score function diverges near the boundary, and simulation around the boundary region becomes infeasible with a numerical SDE solver. To avoid the numerical instability, the simulation is terminated before reaching the boundary point in practice. Moreover, such numerical instability is often observed even during training, especially when the model is trained with a maximum likelihood objective. Therefore, heuristics

like time truncation is widely used in both training and inference of diffusion models. Although time truncation is one of the most naive ways to avoid numerical instability, it requires tuning of the truncation time and also breaks the rigorous formulation of the diffusion model.

In this paper, we propose a method to completely eliminate the heuristic of time truncation from both training and inference of diffusion models. First, to eliminate time truncation during training, we consider sufficient conditions for the maximum likelihood objective not to diverge. Specifically, by using a specific noise schedule and parameterization, we show that the objective becomes always finite even around the boundary points. This prevents the diffusion model from suffering from numerical instability when training with the maximum likelihood objective. We also provide a way to reduce variance of the Monte-Carlo estimate of the objective. Second, we propose a new SDE solver to eliminate time truncation time during sampling. This solver avoids numerical instability at boundary points by taking advantage of the semi-linear structure of the reverse SDE.

By combining these techniques, we successfully remove the dependence on time truncation from both training and inference of the diffusion model. We name this framework *FullDiffusion*. In experiments, we validate the effectiveness of FullDiffusion on CIFAR-10 and ImegeNet 32x32 using DDPM++ as a baseline and confirm that it consistently outperforms the baseline in terms of both likelihood and sample quality measured by the Fréchet inception distance (FID).

## 2 BACKGROUND

### 2.1 DIFFUSION MODELS

In this section, we provide a priliminary knowledge on the concept of diffusion models. Diffusion models are deep generative models that smoothly transform data $\mathbf{x}_0 \in \mathbb{R}^D$ to noise with a diffusion process, and generate samples by learning and simulating the time reversal of this diffusion. First, we consider a following stochastic differential equation to diffuse the data distribution $p_{\text{data}}(\mathbf{x}_0)$ towards a noise distribution (i.e., a standard Gaussian distribution):

$$d\mathbf{x}_t = f_t \mathbf{x}_t dt + g_t d\mathbf{w}, \tag{1}$$

where $f_t$ and $g_t$ are drift and diffusion coefficients, and $\mathbf{w}$ is a standard Wiener process. The solution of an SDE, i.e., $\{\mathbf{x}_t\}_{t \in [0,1]}$, is called a diffusion process. We denote the marginal distribution of $\mathbf{x}_t$ and the transition probability from $\mathbf{x}_0$ to $\mathbf{x}_t$ as $q_t(\mathbf{x}_t)$ and $q_{0t}(\mathbf{x}_t \mid \mathbf{x}_0)$, respectively. In the SDE of Eq. (1), the transition probability $q_{0t}$ can be analytically obtained as follows:

$$q_{0t}(\mathbf{x}_t \mid \mathbf{x}_0) = \mathcal{N}(\mathbf{x}_t; \alpha_t \mathbf{x}_0, \sigma_t^2 \boldsymbol{I}), \tag{2}$$

where $\alpha_t = \exp\left(\int_0^t f_s ds\right)$, and $\sigma_t^2 = \alpha_t^2 \int_0^t \left(g_s^2/\alpha_s^2\right) ds$. By choosing the coefficients $f_t$ and $g_t$ so that $\alpha_1 = 0$ and $\sigma_1 = 1$ hold, the solution of Eq. (1) approaches a standard Gaussian distribution as $t \to 1$, i.e., $q_1(\mathbf{x}_1) = \mathcal{N}(\mathbf{x}_1; \mathbf{0}, \boldsymbol{I})$. There are several ways to meet this condition as listed below[1].

**Variance Preserving (VP)**: When $f_t$ is non-positive and $g_t^2$ is set to $-2f_t$, the SDE is known as the *variance preserving* (VP) SDE, which is widely used for diffusion models. In the VP SDE, $\alpha_t^2 + \sigma_t^2 = 1$ holds. In previous works, $g_t^2$ is often denoted as $\beta_t$ for the VP SDE.

**Sub-VP**: Song et al. (2020b) also propose another type of SDE named sub-VP SDE, in which $g_t^2$ is defined as $-2f_t(1 - e^{\int_0^t 4f_s ds})$. In this case, $\alpha_t^2 + \sigma_t = 1$ holds instead.

**Straight Path (SP)**: When $g_t^2$ is set to $-2f_t(1 - e^{\int_0^t f_s ds})$, the SDE is called the *straight path* (SP) SDE (Zheng et al., 2023), where $\alpha_t + \sigma_t = 1$ holds. The SP SDE is often used for the *optimal transport* (OT) conditional vector field in the context of flow matching (Lipman et al., 2023; Albergo & Vanden-Eijnden, 2023; Liu et al., 2023).

In this paper, we focus on the VP SDE, because it is most widely used in the context of diffusion models (Kingma et al., 2021; Kingma & Gao, 2023). If we can simulate the reverse process of

---

[1]Although the variance exploding (VE) SDE is also widely used, we exclude it here because the VE SDE does not hold $\alpha_1 = 0$ and $\sigma_1 = 1$.

Eq. (1) from a standard Gaussian distribution, we can obtain samples from the data distribution $p_{\text{data}} = q_0$ at $t = 0$. Fortunately, the reverse process of Eq. (1) has an analytical form as follows:

$$d\mathbf{x}_t = \left( f_t \mathbf{x}_t - g_t^2 \mathbf{s}_t \left( \mathbf{x}_t \right) \right) dt + g_t d\bar{\mathbf{w}}, \tag{3}$$

where $\mathbf{s}_t \left( \mathbf{x}_t \right) = \nabla_{\mathbf{x}_t} \log q_t \left( \mathbf{x}_t \right)$, and $\bar{\mathbf{w}}$ is a standard Wiener process in the reverse-time direction. Since this reverse SDE includes a time-dependent score function $\mathbf{s}_t$, which is unknown in advance, we need to estimate it using a parameterized function, such as a neural network, i.e., $\hat{\mathbf{s}}_{\boldsymbol{\theta}} \left( \mathbf{x}_t, t \right) \approx \mathbf{s}_t \left( \mathbf{x}_t \right)$. To fit the function $\hat{\mathbf{s}}_{\boldsymbol{\theta}}$ to the true score function $\mathbf{s}_t$, its parameter $\boldsymbol{\theta}$ is optimized by minimizing the following score matching loss:

$$\mathcal{J}_{\text{SM}} \left( \boldsymbol{\theta} \right) = \frac{1}{2} \mathbb{E} \left[ \lambda_t \left\| \mathbf{s}_t \left( \mathbf{x}_t \right) - \hat{\mathbf{s}}_{\boldsymbol{\theta}} \left( \mathbf{x}_t, t \right) \right\|^2 \right], \tag{4}$$

where $t \sim \mathcal{U} \left( t; 0, 1 \right)$, $\mathbf{x}_t \sim q_t \left( \mathbf{x}_t \right)$, and $\lambda_t$ is some weighting function. Although $\mathcal{J}_{\text{SM}}$ is intractable since the true score $\mathbf{s}_t$ is not accesible, minimization of $\mathcal{J}_{\text{SM}}$ is equivalent to minimization of the following denoising score matching loss (Vincent, 2011):

$$\mathcal{J}_{\text{DSM}} \left( \boldsymbol{\theta} \right) = \frac{1}{2} \mathbb{E} \left[ \lambda_t \left\| \nabla_{\mathbf{x}_t} \log q_{0t} \left( \mathbf{x}_t \mid \mathbf{x}_0 \right) - \hat{\mathbf{s}}_{\boldsymbol{\theta}} \left( \mathbf{x}_t, t \right) \right\|^2 \right] \tag{5}$$

$$= \frac{1}{2} \mathbb{E} \left[ \frac{\lambda_t}{\sigma_t^2} \left\| \boldsymbol{\epsilon} - \hat{\boldsymbol{\epsilon}}_{\boldsymbol{\theta}} \left( \mathbf{x}_t, t \right) \right\|^2 \right], \tag{6}$$

where $\mathbf{x}_0 \sim p_{\text{data}} \left( \mathbf{x}_0 \right)$, $\boldsymbol{\epsilon} \sim \mathcal{N} \left( \boldsymbol{\epsilon}; \mathbf{0}, \boldsymbol{I} \right)$, $\mathbf{x}_t = \alpha_t \mathbf{x}_0 + \sigma_t \boldsymbol{\epsilon}$, and $\hat{\boldsymbol{\epsilon}}_{\boldsymbol{\theta}} \left( \mathbf{x}_t, t \right) = -\sigma_t \hat{\mathbf{s}}_{\boldsymbol{\theta}} \left( \mathbf{x}_t, t \right)$. When $\lambda_t = \sigma_t^2$, the denoising score matching loss $\mathcal{J}_{\text{DSM}}$ is equivalent to a simple noise prediction loss used in the denoising diffusion probabilistic model (DDPM) (Ho et al., 2020) and DDPM++ (Song et al., 2020b). After training, the estimated score function $\hat{\mathbf{s}}_{\boldsymbol{\theta}} \left( \mathbf{x}_t, t \right) = -\hat{\boldsymbol{\epsilon}}_{\boldsymbol{\theta}} \left( \mathbf{x}_t, t \right) / \sigma_t$ is substituted for the true score $\mathbf{s}_t$ to simulate the reverse diffusion process for sample generation:

$$d\mathbf{x}_t = \left( f_t \mathbf{x}_t - g_t^2 \hat{\mathbf{s}}_{\boldsymbol{\theta}} \left( \mathbf{x}_t \right) \right) dt + g_t d\bar{\mathbf{w}} \tag{7}$$

$$= \left( f_t \mathbf{x}_t + \frac{g_t^2}{\sigma_t} \hat{\boldsymbol{\epsilon}}_{\boldsymbol{\theta}} \left( \mathbf{x}_t \right) \right) dt + g_t d\bar{\mathbf{w}}, \tag{8}$$

where $\mathbf{x}_1 \sim p_1 \left( \mathbf{x}_1 \right) = \mathcal{N} \left( \mathbf{x}_1; \mathbf{0}, \boldsymbol{I} \right)$. To simulate the SDE, some numerical sover, e.g., the Euler–Maruyama method (Kloeden et al., 2012), is applied.

## 2.2 Time Truncation in Samping

When simulating the estimated SDE of Eq. (8), we need to confront numerical instability around the boundary points at $t = 0, 1$. For example, when we adopt the VP SDE, the coefficients of Eq. (8) take the following form:

$$f_t = \frac{1}{\alpha_t} \cdot \frac{d\alpha_t}{dt}, \; g_t = -\frac{2}{\alpha_t} \cdot \frac{d\alpha_t}{dt}, \; \frac{g_t^2}{\sigma_t} = -\frac{2}{\alpha_t \sigma_t} \cdot \frac{d\alpha_t}{dt}. \tag{9}$$

Since $\alpha_t \to 0$ as $t \to 1$ and $\sigma_t \to 0$ as $t \to 0$, these coefficients diverge at the boundary points. Therefore, it is difficult to simulate the SDE around $t = 0, 1$ with a naive SDE solver. To avoid the singularity, some heuristics are commonly used in previous works. For instance, Song & Ermon (2019) limit the simulation time within $t \in [t_{\min}, 1]$ instead of $t \in [0, 1]$ to avoid the divergence near $t = 0$. The truncation time $t_{\min}$ is typically set to a small positive number (e.g., $10^{-5}$). In addition, they use a noise schedule such that $g_t^2 = g_{\min}^2 + \left( g_{\max}^2 - g_{\min}^2 \right) t$. In this noise schedule, $\alpha_1$ does not exactly correspond to 0; hence the divergence at $t = 1$ is also avoided, although $q_1 = p_1$ no longer holds. Such heuristics are dominantly used when sampling from continous-time diffusion models after introduced by the original paper by Song & Ermon (2019).

## 2.3 Time Truncation in Maximum Likelihood Training

Song et al. (2021) have shown that when the weighting function $\lambda_t$ in Eq. (6) is equal to $g_t^2$, the denoising score matching loss can be seen as an upper bound of the negative log-likelihood except

for a constant factor as follows:

$$-\mathbb{E}\left[\log p_0\left(\mathbf{x}_0;\boldsymbol{\theta}\right)\right] \leq \mathbb{E}\left[\frac{g_t^2}{2}\left\|\nabla_{\mathbf{x}_t}\log q_{0t}\left(\mathbf{x}_t \mid \mathbf{x}_0\right) - \hat{\mathbf{s}}_{\boldsymbol{\theta}}\left(\mathbf{x}_t, t\right)\right\|^2\right] \tag{10}$$

$$-\mathbb{E}\left[\frac{g_t^2}{2}\left\|\nabla_{\mathbf{x}_t}\log q_{0t}\left(\mathbf{x}_t \mid \mathbf{x}_0\right)\right\|^2 + Df_t + H\left(q_{01}, p_1\right)\right]$$

$$= \mathbb{E}_{\mathbf{x}_0} \mathbb{E}_{t,\boldsymbol{\epsilon}} \underbrace{\left[\frac{g_t^2}{2\sigma_t^2}\left\|\hat{\boldsymbol{\epsilon}}_{\boldsymbol{\theta}}\right\|^2 - \frac{g_t^2}{\sigma_t^2}\hat{\boldsymbol{\epsilon}}_{\boldsymbol{\theta}}^\top \boldsymbol{\epsilon} - Df_t + H\left(q_{01}, p_1\right)\right]}_{\mathcal{L}_{\mathrm{ELBO}}(\mathbf{x}_0, \boldsymbol{\theta})} \tag{11}$$

$$= \mathcal{J}_{\mathrm{DSM}}\left(\boldsymbol{\theta}\right) + \mathrm{const.}, \tag{12}$$

where $\mathbf{x}_0 \sim p_{\mathrm{data}}\left(\mathbf{x}_0\right)$, and $p_t$ is a marginal distribution of the solution of an SDE defined by the estimated score function in Eq. (8). Eq. (12) justifies the minimization of the denoising score matching loss $\mathcal{J}_{\mathrm{DSM}}$ as maximum likelihood training, since it is equivalent to maximizing the evidence lower bound (ELBO).

However, when training diffusion models with the ELBO objective, we again encounter numerical instability around the boundary points, since the coefficients of $\mathcal{L}_{\mathrm{ELBO}}$ include divergent terms. Therefore, heuristics to avoid the singularity, such as time truncation in Sec. 2.2, are also widely used for the maximum likelihood training of diffusion models (Song et al., 2021; Kingma et al., 2021). Song et al. (2021) justifies it by demonstrating that the ELBO objective with time truncation corresponds to maximizing the ELBO for the perturbed data $\mathbf{x}_{t_{\min}} \sim q_{t_{\min}}$ as follows:

$$-\mathbb{E}\left[\log p_{t_{\min}}\left(\mathbf{x}_{t_{\min}};\boldsymbol{\theta}\right)\right] \leq \tilde{\mathcal{J}}_{\mathrm{DSM}}\left(\boldsymbol{\theta}, t_{\min}\right) + \mathrm{const.}, \tag{13}$$

$$\tilde{\mathcal{J}}_{\mathrm{DSM}}\left(\boldsymbol{\theta}, t_{\min}\right) = \int_{t_{\min}}^1 \frac{g_t^2}{2\sigma_t^2}\left\|\boldsymbol{\epsilon} - \hat{\boldsymbol{\epsilon}}_{\boldsymbol{\theta}}\left(\mathbf{x}_t, t\right)\right\|^2 dt. \tag{14}$$

Although the divergence at the boundary points occurs especially for the ELBO objective, time truncation is often used even when training with the non-ELBO objective (e.g., $\lambda_t = \sigma_t^2$ in Song et al. (2020b)).

In summary, the heuristics to avoid the numerical instability at the time boundaries, such as time truncation, are predominantly applied in both trainig and inference time for diffusion models. Althoguh such heuristics help to stabilize training and sampling of diffusion models in practice, they hinder a rigorous correnpondence between the true SDE in Eq. (3) and the estimated SDE in Eq. (8). Furthermore, it is difficult to chooce appropriate values of hyperparameters (e.g., $t_{\min}$, $g_{\min}^2$, and $g_{\max}^2$), requiring additional tuning costs. Our main focus in this paper is to completely eliminate these heuristics without harming the practical performance of the diffusion models.

## 3 METHOD: FULLDIFFUSION

In this section, we provide a way to eliminate the time truncation from both training and sampling of diffusion models. Specifically, we first demonstrate that the divergence of the ELBO objective at the boundary points can be avoided by carefully designing the parameterization and the noise schedule. By this modification, we can eliminate time truncation from training especially for the maximum likelihood objective. Furthermore, we provide a way to reduce the variance of the Monte-Carlo estimation of the ELBO objective using stratified sampling. Finally, to eliminate time truncation from sampling, we introduce a novel numerical SDE solver to avoid the divergence during the SDE simulation. By combining all of them, we can stably train and sample from diffusion models without relying on any heuristics like time truncation. We name the framework of this training and sampling scheme for diffusion models *FullDiffusion*.

### 3.1 PARAMETERIZATION AND NOISE SCHEDULE

As described in the previous section, the negative ELBO, $\mathcal{L}_{\mathrm{ELBO}}$, in Eq. (11) includes divergent coefficients at the time boundaries $t = 0, 1$. This indicates that $\mathcal{L}_{\mathrm{ELBO}}$ almost always diverges to infinity in expectation; hence training is infeasible with the ELBO objective unless relying on time

truncation. However, if the noise predictor $\hat{\epsilon}_{\boldsymbol{\theta}}$ has a structure that nagates the divergence at the time boundaries, the divergence of $\mathcal{L}_{\text{ELBO}}$ can be avoided even when the coefficients are divergent.

More specifically, we derive sufficient conditions regarding the noise schedule and the parameterization to eliminate the divergence as follows:

1. $f_t = -\frac{t}{1-t^2}$ and $g_t = \sqrt{\frac{2t}{1-t^2}}$, which leads to $\alpha_t = \sqrt{1-t^2}$ and $\sigma_t = t$.

2. The noise predictor $\hat{\epsilon}_{\boldsymbol{\theta}}$ takes the following form:

$$\hat{\epsilon}_{\boldsymbol{\theta}}\left(\mathbf{x}_t, t\right) = \sigma_t \left(\sigma_t^2 \mathbf{x}_t - \alpha_t^2 \hat{\boldsymbol{\nu}}_{\boldsymbol{\theta}}\left(\mathbf{x}_t, t\right)\right), \tag{15}$$

where $\hat{\boldsymbol{\nu}}_{\boldsymbol{\theta}}$ is some parametric function defined by a neural network (e.g., U-Net).

Under this parameterization and noise schedule, $\mathcal{L}_{\text{ELBO}}$ takes the following form:

$$\mathcal{L}_{\text{ELBO}}\left(\mathbf{x}_0; \boldsymbol{\theta}\right) = \mathbb{E}\left[\alpha_t \hat{\boldsymbol{\nu}}\left(\mathbf{x}_t, t\right)^{\top}\left(\alpha_t \sigma_t \hat{\boldsymbol{\nu}}\left(\mathbf{x}_t, t\right) + 2\left(\alpha_t\left(1 + \sigma_t^2\right)\boldsymbol{\epsilon} - \sigma_t^3 \mathbf{x}_0\right)\right)\right]$$
$$+ \frac{1}{6}\|\mathbf{x}_0\|^2 + \frac{D}{2}\left(\frac{7}{6} + \log\left(2\pi\right)\right). \tag{16}$$

The derivation is provided in Appendix A. It can be seen that the divergent coefficients are eliminated from $\mathcal{L}_{\text{ELBO}}$ under this difinition; hence diffusion models can be trained with this objective without relying on time truncation. In addition, the boundary conditions, i.e., $(\alpha_0, \sigma_0) = (1, 0)$ and $(\alpha_1, \sigma_1) = (0, 1)$, strictly hold for this noise schedule, so this definition does not break the correspondence between the true SDE and the estimated SDE.

In fact, this parameterization of the noise predictor $\hat{\epsilon}_{\boldsymbol{\theta}}$ is a very natural choice when we see it as an estimator of the score function. Under this definition of $\hat{\epsilon}_{\boldsymbol{\theta}}$, the estimated score function $\hat{\mathbf{s}}_{\boldsymbol{\theta}} = -\hat{\epsilon}_{\boldsymbol{\theta}}/\sigma_t$ has the following form:

$$\hat{\mathbf{s}}_{\boldsymbol{\theta}}\left(\mathbf{x}_t, t\right) = \alpha_t^2 \hat{\boldsymbol{\nu}}_{\boldsymbol{\theta}}\left(\mathbf{x}_t, t\right) - \sigma_t^2 \mathbf{x}_t \tag{17}$$

When the time $t$ approaches 1, this score estimator converges to $-\mathbf{x}_1$, which corresponds to the score function of the standard Gaussian distribution, whereas it converges to $\hat{\boldsymbol{\nu}}_{\boldsymbol{\theta}}\left(\mathbf{x}_0, 0\right)$ as $t \to 0$. Therefore, the neural network $\hat{\boldsymbol{\nu}}_{\boldsymbol{\theta}}\left(\cdot, t\right)$ will naturally learn the interpolation between the score function of the non-perturbed data $\mathbf{x}_0$ and the one of the pure Gaussian distribution of $\mathbf{x}_1$ by definition.

### 3.2 VARIANCE REDUCTION VIA STRATIFIED SAMPLING

So far, we have focused on a way to fix the divergence of the ELBO itself. However, to train diffusion models in a feasible manner, the variance of the Monte Carlo estimate of the ELBO should also be small. Song et al. (2021) propose to use importance weighting to reduce the variance of the maximum likelihood objective, but it cannot be directly applied to our case due to the difference of the parameterization. Instead, we propose to use stratified sampling for the time variable $t$ for variance reduction. When we estimate the expectation of the ELBO over the training set using a minbatch of $n$ data $\left\{\mathbf{x}_0^{(i)}\right\}_{i=1}^{n}$, we construct an unbiased estimator of the expectation as follows:

$$\mathbb{E}_{\mathbf{x}_0}\left[\mathcal{L}_{\text{ELBO}}\left(\mathbf{x}_0; \boldsymbol{\theta}\right)\right]$$
$$= \mathbb{E}\left[\frac{1}{n}\sum_{i=1}^{n}\alpha_{t_i}\hat{\boldsymbol{\nu}}_{\boldsymbol{\theta}}\left(\mathbf{x}_{t_i}^{(i)}, t_i\right)^{\top}\left(\alpha_{t_i}\sigma_{t_i}\hat{\boldsymbol{\nu}}_{\boldsymbol{\theta}}\left(\mathbf{x}_{t_i}^{(i)}, t_i\right) + 2\left(\alpha_{t_i}\left(1 + \sigma_{t_i}^2\right)\boldsymbol{\epsilon} - \sigma_{t_i}^3 \mathbf{x}_0^{(i)}\right)\right)\right]$$
$$+ \frac{1}{6n}\sum_{i=1}^{n}\left\|\mathbf{x}_0^{(i)}\right\|^2 + \frac{D}{2}\left(\frac{7}{6} + \log\left(2\pi\right)\right), \tag{18}$$

where $t_i \sim \mathcal{U}\left(t_i; (i-1)/n, i/n\right)$. We experimentally observe that this technique is effective to reduce the variance of the Monte-Carlo estimation and stabilize the training.

### 3.3 FULLDIFFUSION-SOLVER: A SPECIAL SDE SOLVER FOR FULLDIFFUSION

Under our parameterization, the reverse-time diffusion in Eq. (8) takes the following form:

$$d\mathbf{x}_t = -t\left(\frac{1 - 2t^2}{1 - t^2}\mathbf{x}_t + 2\hat{\boldsymbol{\nu}}_{\boldsymbol{\theta}}\left(\mathbf{x}_t, t\right)\right)dt + \sqrt{\frac{2t}{1 - t^2}}d\bar{\mathbf{w}} \tag{19}$$

---

**Algorithm 1** FullDiffusion-Solver-1

---

**Require:** Number of discritization steps $M$, Predictor $\hat{\boldsymbol{\nu}}_{\boldsymbol{\theta}}$

$\quad \mathbf{x}_s \sim \mathcal{N}\left(\mathbf{x}_s; \mathbf{0}, \boldsymbol{I}\right)$

$\quad s \leftarrow 1$

$\quad$**for** $i \leftarrow 1$ to $M$ **do**

$\quad\quad t \leftarrow s - 1/M$

$\quad\quad \mathbf{x}_t \sim \mathcal{N}\left(\mathbf{x}_t; \sqrt{\frac{1-s^2}{1-t^2}}\left(\left(1+s^2-t^2\right)\mathbf{x}_s + \left(s^2-t^2\right)\hat{\boldsymbol{\nu}}_{\boldsymbol{\theta}}\left(\mathbf{x}_s, s\right)\right), \frac{t^2\left(s^2-t^2\right)}{s^2\left(1-t^2\right)}\boldsymbol{I}\right)$

$\quad\quad s \leftarrow t,\ \mathbf{x}_s \leftarrow \mathbf{x}_t$

$\quad$**end for**

$\quad$**return** $\mathbf{x}_t$

---

Since the coefficients of the first and last terms diverges at $t = 1$, it is still difficult to simulate it using a naive SDE solver, such as the Euler–Maruyama method. However, we can avoid the singularity by utilizing the semi-linear structure of the SDE as proposed by Lu et al. (2022a;b). First, we reformulate the SDE with the signal predictor $\hat{\mathbf{x}}_{\boldsymbol{\theta}}$ as follows:

$$d\mathbf{x}_t = \frac{1}{t}\left(\frac{2-t^2}{1-t^2}\mathbf{x}_t - \frac{2}{\sqrt{1-t^2}}\hat{\mathbf{x}}_{\boldsymbol{\theta}}\left(\mathbf{x}_t, t\right)\right)dt + \sqrt{\frac{2t}{1-t^2}}d\bar{\mathbf{w}}, \tag{20}$$

$$\text{where } \hat{\mathbf{x}}_{\boldsymbol{\theta}}\left(\mathbf{x}_t, t\right) = \left(\mathbf{x}_t - \hat{\boldsymbol{\epsilon}}_{\boldsymbol{\theta}}\left(\mathbf{x}_t, t\right)\right)/\alpha_t$$

$$= \alpha_t\left(\left(1+\sigma_t^2\right)\mathbf{x}_t + \sigma_t^2\hat{\boldsymbol{\nu}}_{\boldsymbol{\theta}}\left(\mathbf{x}_t, t\right)\right) \tag{21}$$

The solution for this SDE given the initial state $\mathbf{x}_s$ can be analytically derived as follows:

$$\mathbf{x}_t = e^{\int_s^t \frac{2-u^2}{u(1-u^2)}du}\mathbf{x}_s - \int_s^t \frac{2e^{\int_\tau^t \frac{2-u^2}{u(1-u^2)}du}}{\tau\sqrt{1-\tau^2}}\hat{\mathbf{x}}_{\boldsymbol{\theta}}\left(\mathbf{x}_\tau, \tau\right)d\tau + \int_s^t \sqrt{\frac{2\tau}{1-\tau^2}}e^{\int_\tau^t \frac{2-u^2}{u(1-u^2)}du}d\mathbf{w}_\tau \tag{22}$$

$$= \frac{\alpha_s\sigma_t^2}{\alpha_t\sigma_s^2}\mathbf{x}_s - \frac{2\sigma_t^2}{\alpha_t}\int_s^t \frac{1}{\sigma_\tau^3}\hat{\mathbf{x}}_{\boldsymbol{\theta}}\left(\mathbf{x}_\tau, \tau\right)d\tau + \frac{\sqrt{2}\sigma_t^2}{\alpha_t}\int_s^t \sigma_\tau^{-3/2}d\mathbf{w}_\tau, \tag{23}$$

where $0 \le t < s \le 1$. Using a first-order approximation for the second term, we can derive a first-order solver for the SDE:

$$\mathbf{x}_t \approx \frac{t^2\sqrt{1-s^2}}{s^2\sqrt{1-t^2}}\mathbf{x}_s + \frac{s^2-t^2}{s^2\sqrt{1-t^2}}\hat{\mathbf{x}}_{\boldsymbol{\theta}}\left(\mathbf{x}_s, s\right) + \frac{t\sqrt{s^2-t^2}}{s\sqrt{1-t^2}}\boldsymbol{\xi} \tag{24}$$

$$= \sqrt{\frac{1-s^2}{1-t^2}}\left(\left(1+s^2-t^2\right)\mathbf{x}_s + \left(s^2-t^2\right)\hat{\boldsymbol{\nu}}_{\boldsymbol{\theta}}\left(\mathbf{x}_s, s\right)\right) + \frac{t}{s}\sqrt{\frac{s^2-t^2}{1-t^2}}\boldsymbol{\xi} \tag{25}$$

$$\coloneqq \tilde{\mathbf{x}}_t \tag{26}$$

where $\boldsymbol{\xi} \sim \mathcal{N}\left(\boldsymbol{\xi}; \mathbf{0}, \boldsymbol{I}\right)$. Since $s > 0$ and $t < 1$ always hold, this solver does not suffer from the divergence at all timesteps; hence it can be applied without relying on time truncation.

Furthermore, we can extend it to a second-order approximation using the Runge–Kutta (RK) method (Runge, 1895; Kutta, 1901; Rößler, 2009) as follows:

$$\mathbf{x}_t \approx \frac{t^2\sqrt{1-s^2}}{s^2\sqrt{1-t^2}}\mathbf{x}_s + \frac{s^2-t^2}{s^2\sqrt{1-t^2}}\left(\left(1-\frac{1}{2c}\right)\hat{\mathbf{x}}_{\boldsymbol{\theta}}\left(\mathbf{x}_s, s\right) + \frac{1}{2c}\hat{\mathbf{x}}_{\boldsymbol{\theta}}\left(\tilde{\mathbf{x}}_r, r\right)\right) + \frac{t\sqrt{s^2-t^2}}{s\sqrt{1-t^2}}\boldsymbol{\xi} \tag{27}$$

$$= \tilde{\mathbf{x}}_t + \frac{s^2-t^2}{2cs^2\sqrt{1-t^2}}\left(\hat{\mathbf{x}}_{\boldsymbol{\theta}}\left(\tilde{\mathbf{x}}_r, r\right) - \hat{\mathbf{x}}_{\boldsymbol{\theta}}\left(\mathbf{x}_s, s\right)\right), \tag{28}$$

where $0 < c \le 1$, $r = s + c\left(t - s\right)$. We set $c = 2/3$, which is known as the Ralston's method (Ralston, 1962) that has the smallest local approximation error among two-stage RK methods. The algorithms of our solvers are summarized in Algorithms 1 and 2. We name our first- and second-order solvers *FullDiffusion-Solver-1* and *-2*, respectively.

As Song et al. (2020b) pointed out, there exists a corresponding probability flow ODE that shares the same marginal density with the forward SDE in Eq. (1).

$$d\mathbf{x}_t = \left(f_t\mathbf{x}_t - \frac{1}{2}g_t^2\mathbf{s}_t\left(\mathbf{x}_t\right)\right)dt \tag{29}$$

---

**Algorithm 2** FullDiffusion-Solver-2

---

**Require:** Number of discritization steps $M$, Predictor $\hat{\boldsymbol{\nu}}_{\boldsymbol{\theta}}$

   $\mathbf{x}_s \sim \mathcal{N}\left(\mathbf{x}_s; \mathbf{0}, \boldsymbol{I}\right)$

   $s \leftarrow 1$

   **for** $i \leftarrow 1$ to $M$ **do**

      $t \leftarrow s - 1/M, \; r \leftarrow s - 2/\left(3M\right)$

      $\boldsymbol{\xi} \sim \mathcal{N}\left(\boldsymbol{\xi}; \mathbf{0}, \boldsymbol{I}\right)$

      $\hat{\boldsymbol{\nu}}_s \leftarrow \hat{\boldsymbol{\nu}}_{\boldsymbol{\theta}}\left(\mathbf{x}_s, s\right)$

      $\tilde{\mathbf{x}}_r \leftarrow \sqrt{\frac{1-s^2}{1-r^2}}\left(\left(1+s^2-r^2\right)\mathbf{x}_s + \left(s^2-r^2\right)\hat{\boldsymbol{\nu}}_s\right) + \frac{r}{s}\sqrt{\frac{s^2-r^2}{1-r^2}}\boldsymbol{\xi}$

      $\tilde{\mathbf{x}}_t \leftarrow \sqrt{\frac{1-s^2}{1-t^2}}\left(\left(1+s^2-t^2\right)\mathbf{x}_s + \left(s^2-t^2\right)\hat{\boldsymbol{\nu}}_s\right) + \frac{t}{s}\sqrt{\frac{s^2-t^2}{1-t^2}}\boldsymbol{\xi}$

      $\hat{\boldsymbol{\nu}}_r \leftarrow \hat{\boldsymbol{\nu}}_{\boldsymbol{\theta}}\left(\tilde{\mathbf{x}}_r, r\right)$

      $\hat{\mathbf{x}}_s \leftarrow \sqrt{1-s^2}\left(\left(1+s^2\right)\mathbf{x}_s + s^2\hat{\boldsymbol{\nu}}_s\right), \; \hat{\mathbf{x}}_r \leftarrow \sqrt{1-r^2}\left(\left(1+r^2\right)\tilde{\mathbf{x}}_r + r^2\hat{\boldsymbol{\nu}}_r\right)$

      $\mathbf{x}_t \leftarrow \tilde{\mathbf{x}}_t + \frac{3\left(s^2-t^2\right)}{4s^2\sqrt{1-t^2}}\left(\hat{\mathbf{x}}_r - \hat{\mathbf{x}}_s\right)$

      $s \leftarrow t, \; \mathbf{x}_s \leftarrow \mathbf{x}_t$

   **end for**

   **return** $\mathbf{x}_t$

---

By approximating the score function $\mathbf{s}_t\left(\cdot\right)$ with the estimator $\hat{\mathbf{s}}_{\boldsymbol{\theta}}\left(\cdot, t\right) = -\hat{\boldsymbol{\epsilon}}_{\boldsymbol{\theta}}\left(\cdot, t\right)/\sigma_t$, the ODE takes the following simple form under the noise schedule and the parameterization in Section 3.1:

$$d\mathbf{x}_t = -\sigma_t\left(\mathbf{x}_t + \hat{\boldsymbol{\nu}}_{\boldsymbol{\theta}}\left(\mathbf{x}_t, t\right)\right)dt \tag{30}$$

Therefore, when using an ODE sampler, we do not need to care about the numerical instability, and can use any sampler, such as the Euler method, the Heun's method and so forth. In addition, we can evaluate the exact likelihood of the ODE via the instantaneous change of variables formula as proposed in Song et al. (2020b).

## 4 RELATED WORKS

### 4.1 NUMERICAL INSTABILITY IN DIFFUSION MODELS

The numerical instability of continuous-time diffusion models around the boundary points has been widely recognized ever since the original paper by Song & Ermon (2019). However, to the best of our knowledge, almost all previous works still rely on time truncation to deal with it (Kingma et al., 2021; Karras et al., 2022). One of the most related attempts regarding this topic is a technique called *soft truncation* (Kim et al., 2022), in which the truncation time $t_{\min}$ is randomly chosen during training. Although soft truncation alleviates the numerical instablity during training, it still requires the choice of a minimum truncation time. Yang et al. (2024) have also tackled the issue of the numerical instability, and pointed out that the Lipschitz constant of the noise predictor $\hat{\epsilon}_{\boldsymbol{\theta}}$ tends to diverge near the boundary point at $t = 0$. To alleviate it, they propose to round the time variable $t$ near the boundary point with a staircase function when inputting small $t$ to the noise predictor. While they experimentally demonstrate the effectiveness of this method, they only apply it to the discrete-time diffusion model, so the applicability to the continuous-time model is still unclear. Moreover, the rounding operation loses information about time near the boundary point, which may leads to performance degradation especially for continuous-time models. On the other hand, our method can fundamentally solve the problem of numerical instability by the design of the model parameterization, the noise schedule, and the numerical solver.

### 4.2 MAXIMUM LIKELIHOOD TRAINING OF DIFFUSION MODELS

Originally, Ho et al. (2020) derived an ELBO objective for the discrete-time diffusion model, but they experimentally show that a non-ELBO objective performs better in terms of the sample quality. After Song et al. (2020b) reformulate the continuous-time diffusion model using stochastic differential equations, Song et al. (2021) and Huang et al. (2021) derive the corresponding ELBO objective for it. In previous works, it is reported that the ELBO objective tends to perform better in terms of

Table 1: Negative log-likelihood (bits/dim) and sample quality (FID scores) on CIFAR-10 and ImageNet $32 \times 32$. Bold indicates best result in the corresponding column. Lower is better.

| Model | | CIFAR-10 | | | | ImageNet $32 \times 32$ | | | |
|---|---|---|---|---|---|---|---|---|---|
| | | NLL | | FID | | NLL | | FID | |
| | $t_{\min}$ | SDE | ODE | SDE | ODE | SDE | ODE | SDE | ODE |
| Baseline | $10^{-5}$ | $\leq 3.28$ | 3.16 | 2.55 | 3.98 | $\leq 3.62$ | 3.56 | 5.42 | 5.68 |
| + ELBO loss | $10^{-5}$ | $\leq 3.08$ | 2.95 | 5.87 | 6.03 | $\leq 3.61$ | 3.55 | 11.15 | 14.14 |
| FullDiffusion | **0** | $\leq \mathbf{2.83}$ | **2.80** | **2.53** | **2.89** | $\leq \mathbf{3.41}$ | **3.41** | **5.00** | **5.02** |
| − Var. reduction | **0** | $\leq 2.86$ | 2.85 | 2.58 | 2.92 | $\leq 3.50$ | 3.48 | 5.13 | 5.18 |

the likelihood evaluation, but the sample quality is likely to degrade compared to the simple noise prediction loss (i.e., $\lambda_t = \sigma_t^2$). However, we experimentally observe that, when using our method, the ELBO objective shows good performance in terms of both likelihood and sample quality, which will be shown in Section 5.

### 4.3 PARAMETERIZATION & NOISE SCHEDULE

In the original paper by Song & Ermon (2019), the noise predictor $\hat{\epsilon}_{\boldsymbol{\theta}}$ is directly parameterized by a neural network (e.g., U-Net), and many subsequent works follow that parameterization. However, some variants are also proposed in the previous works, such as the signal predictor $\hat{\mathbf{x}}_{\boldsymbol{\theta}} = (\mathbf{x}_t - \hat{\epsilon}_{\boldsymbol{\theta}}) / \alpha_t$, the velocity predictor $\hat{\mathbf{v}}_{\boldsymbol{\theta}} = (\hat{\epsilon}_{\boldsymbol{\theta}} - \sigma_t \mathbf{x}_t) / \alpha_t$ (Salimans & Ho, 2022). However, these variants also suffer from the numerical instability around the boundary points, so they do not contribute to our motivation.

On the noise schedule, Song & Ermon (2019) use the linear $g_t^2$ schedule as described in Section 2.2, but many variants have been proposed in previous works. For example, the cosine $\alpha_t$ schedule is often used (Nichol & Dhariwal, 2021; Salimans & Ho, 2022; Choi et al., 2022). In this paper, we show that the combination of the linear $\sigma_t$ schedule and the parameterization in Eq. (15) contributes to the stable maximum likelihood training without time truncation. However, there might be other variants to achive the same goal, which we leave as future work.

## 5 EXPERIMENT

To demonstrate the effectiveness of our FullDiffusion, we perform experiments of image generation and density estimation tasks. We use DDPM++ (Song & Ermon, 2019) for VP SDE as a baseline model, and perform an ablation study by modifying the design of parameterization, noise schedule, and numerical solvers as explained in Section 3. We also compare with DDPM++ trained with the ELBO objective as proposed in Song et al. (2021). Our experimental settings are based on the original papers by Song et al. (2020b; 2021), and our implementations are also based on their official codes.

**Datasets:** In our experiment, we use the CIFAR-10 and downsampled ImageNet (Deng et al., 2009) datasets. Note that the old version of the downsampled ImageNet dataset used in Song et al. (2021) is no longer available, so we adopt the new version of $32 \times 32$ resolution images provided at `https://image-net.org`. For fair comparison, we reimplement the official codes of Song et al. (2021) for the new version of the downsampled ImageNet dataset, and compare the performance under the same settings. Following the setting of Song & Ermon (2019); Song et al. (2021), we use uniform dequantization to map the 8-bit images into a continuous space, since diffusion models are designed for continuous data. We did not adopt variational dequantization in this experiment.

**Evaluation:** We evalutate the model performance with the negative log-liklihooed of the reverse SDE and the probability flow ODE, and the Fréchet inception distance (FID) of the generated images via SDE/ODE samplers. Since the negative log-likelihood for the reverse SDE is intractable, we report its upper bound as in Song et al. (2021). We use FullDiffusion-Solver-2 introduced in Section

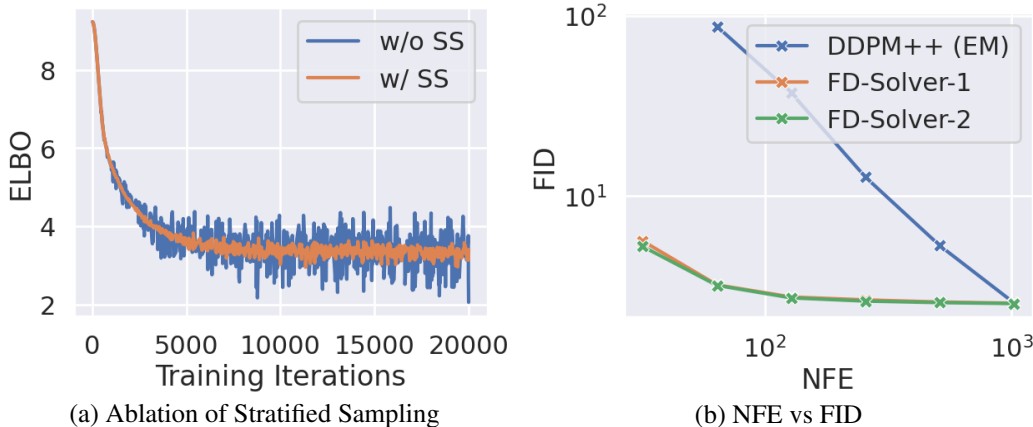

(a) Ablation of Stratified Sampling        (b) NFE vs FID

Figure 1: (a) Training curves of FullDiffusion with/without stratified sampling (SS). (b) Sample quality measured by FID of the baseline model (DDPM++) and FullDiffusion (FD) for CIFAR-10. For the baseline model, the Euler–Maruyama (EM) solver is applied.

3.3 and Euler method as SDE and ODE solvers respectively in order to generate samples for FID evaluation of the FullDiffusion model.

The result is summarized in Table 1. Our key observations are as follows:

- FullDiffusion consistently performs beter than the baseline models in terms of both the test likelihood and the FID, whereas previous studies have reported that there is a trade-off between likelihood and FID, and models trained with an ELBO objective tend to perform poorer in terms of FID. This may be due to the fact that our method eliminates numerical instability in maximum likelihood training and sampling.

- FullDiffusion tends to have small gaps between SDE and ODE in the likelihood evaluation. This indicates that FullDiffusion has a tight variational lower bound.

- Variance reduction via stratified sampling slightly improves the overall performance in terms of both the likelihood and the FID.

**Effect of variance reduction:** To demonstrate the effectiveness of the stratified sampling for variance reduction of the Monte-Carlo estimate, we provide learning curves when training FullDiffusion for CIFAR-10 with or without the stratified sampling in Figure 1 (a). It can be seen that the loss variance is significantly reduced by introducing stratified sampling. Although the variance is relatively small even without stratified sampling, FullDiffusion can be trained more stably by using it.

**Performance of FullDiffusion-Solvers:** We also compare the performance of our first- and second-order FullDiffusion-Solvers in terms of sample quality measured by FID scores for CIFAR-10. We vary different number of function evaluations (NFE) which is the numebr of calls to the model $\hat{\nu}_\theta$. The results are shown in Figure 1 (b). We observe that the FID converges to good sample quality around 100 NFE even with the first-order solver, and the convergence accelerates slightly by using the second-order solver, whereas the original DDPM++ requires about 1,000 NFE with the Euler–Maruyama method to reach good quality. This indicates that our FullDiffusion-Solvers are effective not only to avoid the divergence at the boundary points but also to efficiently generate samples compared to naive solvers (e.g., the Euler–Maruyama method). The generated samples of CIFAR-10 by our FullDiffusion-Solver-2 are visualized in Figure 2.

## 6 CONCLUSION

In this paper, we propose FullDiffusion, a framework to train and infer score-based diffusion models without relying on time truncation around the boundary points. To overcome inherent numerical instability of diffusion models, we reformulate the parameterization and the noise schedule so that the maximum likelihood objective does not diverge around the boundary points. Moreover, to avoid the divergence during SDE simulation, we propose a special SDE solver named FullDiffusion-Solver.

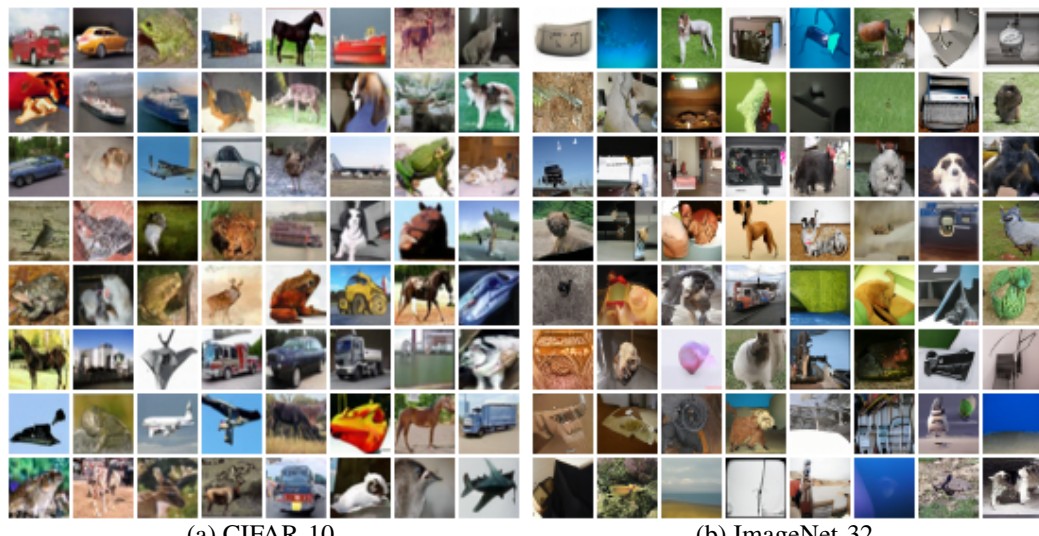

(a) CIFAR-10         (b) ImageNet-32

Figure 2: Generated samples of (a) CIFAR-10 and (b) ImageNet-32 by FullDiffusion-Solver-2.

By combining these techniques, we completely eliminate heuristics like time truncation to alleviate the numerical instability from continuouse-time diffusion models. We experimentally observe that our FullDiffusion consistently outperforms the baseline models in terms of both likelihood evaluation and sample quality measured by FID scores. Our experiments only include low-resolution image generation, such as CIFAR-10, so validation in more large-scale and high-resolution datasets is promising future direction. We hope that this work will help practioners eliminate troublesome hyperparameter tunings regarding numerical instability (e.g., truncation time $t_{\min}$) of diffusion models.

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

## A  DERIVATION OF EQ. (16)

$$\mathcal{L}_{\text{ELBO}}(\mathbf{x}_0, \boldsymbol{\theta}) = \mathbb{E}\left[\frac{g_t^2}{2\sigma_t^2}\|\hat{\boldsymbol{\epsilon}}_{\boldsymbol{\theta}}\|^2 - \frac{g_t^2}{\sigma_t^2}\hat{\boldsymbol{\epsilon}}_{\boldsymbol{\theta}}^\top \boldsymbol{\epsilon} - Df_t + H(q_{01}, p_1)\right] \tag{31}$$

$$= \mathbb{E}\left[\frac{1}{\alpha_t^2 \sigma_t}\hat{\boldsymbol{\epsilon}}_{\boldsymbol{\theta}}^\top (\hat{\boldsymbol{\epsilon}}_{\boldsymbol{\theta}} - 2\boldsymbol{\epsilon}) + \frac{D\sigma_t}{\alpha_t^2}\right] + \frac{D}{2}(1 + \log(2\pi)) \tag{32}$$

$$= \mathbb{E}\left[\frac{1}{\alpha_t^2}\left(\sigma_t^2 \mathbf{x}_t - \alpha_t^2 \hat{\boldsymbol{\nu}}_{\boldsymbol{\theta}}\right)^\top \left(\sigma_t \left(\sigma_t^2 \mathbf{x}_t - \alpha_t^2 \hat{\boldsymbol{\nu}}_{\boldsymbol{\theta}}\right) - 2\boldsymbol{\epsilon}\right) + \frac{D\sigma_t}{\alpha_t^2}\right]$$
$$+ \frac{D}{2}(1 + \log(2\pi)) \tag{33}$$

$$= \mathbb{E}\left[\frac{1}{\alpha_t^2}\left(\alpha_t \sigma_t^2 \mathbf{x}_0 - \alpha_t^2 \hat{\boldsymbol{\nu}}_{\boldsymbol{\theta}} + \sigma_t^3 \boldsymbol{\epsilon}\right)^\top \left(\alpha_t \sigma_t^3 \mathbf{x}_0 - \alpha_t^2 \sigma_t \hat{\boldsymbol{\nu}}_{\boldsymbol{\theta}} + \left(\sigma_t^4 - 2\right)\boldsymbol{\epsilon}\right) + \frac{D\sigma_t}{\alpha_t^2}\right]$$
$$+ \frac{D}{2}(1 + \log(2\pi)) \tag{34}$$

$$= \mathbb{E}\left[\sigma_t \left\|\sigma_t^2 \mathbf{x}_0 - \alpha_t \hat{\boldsymbol{\nu}}_{\boldsymbol{\theta}}\right\|^2 + 2\alpha_t^2 \left(1 + \sigma_t^2\right)\boldsymbol{\epsilon}^\top \hat{\boldsymbol{\nu}}_{\boldsymbol{\theta}} + D\sigma_t \left(\sigma_t^4 + \sigma_t^2 - 1\right)\right]$$
$$+ \frac{D}{2}(1 + \log(2\pi)) \tag{35}$$

$$= \mathbb{E}\left[\alpha_t^2 \sigma_t \left\|\hat{\boldsymbol{\nu}}_{\boldsymbol{\theta}}\right\|^2 - 2\alpha_t \sigma_t^3 \hat{\boldsymbol{\nu}}_{\boldsymbol{\theta}}^\top \mathbf{x}_0 + 2\alpha_t^2 \left(1 + \sigma_t^2\right)\boldsymbol{\epsilon}^\top \hat{\boldsymbol{\nu}}_{\boldsymbol{\theta}}\right]$$
$$+ \mathbb{E}\left[\sigma_t^5 \|\mathbf{x}_0\|^2 - D\sigma_t \left(\sigma_t^4 + \sigma_t^2 - 1\right)\right] + \frac{D}{2}(1 + \log(2\pi)) \tag{36}$$

$$= \mathbb{E}\left[\alpha_t^2 \sigma_t \left\|\hat{\boldsymbol{\nu}}_{\boldsymbol{\theta}}\right\|^2 - 2\alpha_t \sigma_t^3 \hat{\boldsymbol{\nu}}_{\boldsymbol{\theta}}^\top \mathbf{x}_0 + 2\alpha_t^2 \left(1 + \sigma_t^2\right)\boldsymbol{\epsilon}^\top \hat{\boldsymbol{\nu}}_{\boldsymbol{\theta}}\right]$$
$$+ \frac{1}{6}\|\mathbf{x}_0\|^2 + \frac{D}{2}\left(\frac{7}{6} + \log(2\pi)\right). \tag{37}$$

For the derivation, we have used the following facts:

$$\mathbb{E}\left[\|\boldsymbol{\epsilon}\|^2\right] = D, \mathbb{E}\left[\boldsymbol{\epsilon}^\top \mathbf{x}_0\right] = 0. \tag{38}$$

## B  DETAILS OF EXPERIMENTAL SETUPS

### B.1  CODE

Our implementation for the experiment is available at `https://anonymous.4open.science/r/fulldiffusion_iclr2025-54A1/`.

### B.2  TOTAL AMOUNT OF COMPUTE

We run our experiments mainly on cloud GPU instances with $8\times$ A100. It took approximately 330 hours for our experiments in total.

### B.3  LICENSE OF ASSETS

**Datasets:** The terms of access for the CIFAR-10 database is provided at `https://www.cs.toronto.edu/~kriz/cifar.html` The terms of access for the ImageNet database is provided at `https://www.image-net.org/download`.

**Code:** Our implementation is based on the official PyTorch code of Song et al. (2020b) provided at `https://github.com/yang-song/score_sde_pytorch/tree/main`.

## C  APPENDIX

You may include other additional sections here.

