# OpenReview forum: "FullDiffusion: Diffusion Models Without Time Truncation"
_ICLR.cc/2025/Conference — Submitted to ICLR 2025_

### Official Review · Reviewer_tN83 · 2024-10-23

**Soundness:** 2
**Presentation:** 1
**Contribution:** 2
**Rating:** 5
**Confidence:** 4

**Summary:**

This paper focuses on the time truncation parameter that causes the divergent score function in diffusion models. To remove the time truncation, the authors propose FullDiffusion by reparametrizing the network prediction and the noise schedule. Under this new parameterization, the authors accordingly propose a first-order solver and a second-order solver inspired by the semi-linear structure of the reverse SDE (DPM-solver). Results on Cifar10 and ImageNet32 show that FullDiffusion outperforms DDPM++ in terms of FID and likelihood.

**Strengths:**

1. The reparameterization of network prediction and noise schedule is novel, which eliminates the singularity issue of time truncation
2. The corresponding solvers are derived along with the new parameterization.
3. FullDiffusion achieves improvements on both FID and NLL

**Weaknesses:**

1. My first major concern is that, the time truncation might be not a problem according to the good FID and NLL achieved by VDM [2] and SoftTruncation [3] (even better than FullDiffusion), these two models maintain the time truncation. Although people know that time truncation causes numerical instability, [1] and [2] proposed different time sampling methods to stabilize the training. Also, I do not think researchers tune the truncation parameter anymore since it is already found and often used as a fixed parameter.

2. The key section 3.1 is ambiguous, e.g. why directly set $\sigma_t=t$, $f_t=-t/(1-t^2)$ and what gives eq 15? I guess the authors want to eliminate the divergent coefficients and these parameterizations are derived from this goal? However, the reasoning, motivations, and derivation are missing in this section.

3. In the abstract, the authors say 'our method eliminates numerical instability during training', but why is there still a big ELBO variance during training (see Figure 2a)? What is the motivation for using stratified sampling to reduce the training variance?

4. This paper excludes the VE-SDE which is also widely used in the community.

5. Benchmarking on only cifar10 and ImageNet32 is insufficient, I suggest the authors test the method on celeba64 and ImageNet64. Also, the authors should compare FullDiffusion with other diffusion models focusing on likelihood, e.g. VDM [2] and SoftTruncation [3]

6. The FID improvement of FullDiffusion is limited, e.g. 5.42-->5.00, 2.55-->2.53. These improvements can even be derived by using different batches of generated samples.

7. The literature review (section 4.2) is insufficient, lacks reviews of major papers, like [2] and [3]

Others:
1) in eq 11, the notations D and H are used without definition.
2) line 402, the velocity predictor looks wrong, according to [4]


[1] Song, Yang, et al. "Maximum likelihood training of score-based diffusion models." NIPS, 2021.

[2] Kingma, Diederik, et al. "Variational diffusion models." NIPS, 2021

[3] Kim, Dongjun, et al. "Soft truncation: A universal training technique of score-based diffusion model for high precision score estimation." ICML, 2022.

[4] Tim Salimans and Jonathan Ho. Progressive distillation for fast sampling of diffusion models. ICLR, 2022

**Questions:**

I am surprised to see that DDPM++ requires 1000 NFE with Euler solver to reach near optimal FID (figure 1b). iDDPM [5] shows that 100-300 NFE can achieves sub-optimal FID by changing the noise schedule. Since FullDiffusion uses a different noise schedule from DDPM++, I am curious about how much the sampling efficiency of FullDiffusion is contributed by the noise schedule.


[5] Nichol, Alexander Quinn, and Prafulla Dhariwal. "Improved denoising diffusion probabilistic models." ICML, 2021.

---

### Official Review · Reviewer_BNou · 2024-10-29

**Soundness:** 3
**Presentation:** 2
**Contribution:** 3
**Rating:** 5
**Confidence:** 3

**Summary:**

This paper mainly considers to remove time-truncation when performing training and sampling of diffusion models.  The main contribution is to propose a new form of the estimated Gaussian noise. As a result, the corresponding LELB bound is nicely defined at the boundary point. A new semi-linear SDE solver is proposed accordingly.

**Strengths:**

(1) A new form of the noise predictor in diffusion models is proposed in order for the LELB bound to be well defined at the bound points (i.e., t=0 and t=1). By doing so, time truncation can be avoided, which I think is nice.

(2) One interesting result is that the FID scores of the ODE solver and SDE solvers are very close in the paper. This suggests that it might be because of the time truncation in the literature, that leads to the poor performance of ODE solver in comparison to SDE solver.

**Weaknesses:**

(1) It is not clear to me how stratified sampling is implemented by reading Section 3.2.  The authors only state that "we propose to use stratified sampling for the time variable t for variance reduction." without providing implementation details.

(2) Is Equation (18) the objective function to be minimized? If so, the authors should explicitly say it. The authors should also elaborate the training time and the GPU they used in their experiments. The link to the source code is empty.

(3) It is not clear how many timesteps are used in Table 1.

(4) The English language needs to be improved. There are quite a few typos in the paper, such as "priliminary", "Althoguh", "after introduced by the original paper by X", "difinition", and "eliminate time truncation time during sampling".

**Questions:**

(1) The authors provided a link for their source code in the appendix. But it is empty when I try to study and re-run their code.

**Details Of Ethics Concerns:**

n. a.

---

### Official Review · Reviewer_6csB · 2024-11-01

**Soundness:** 2
**Presentation:** 2
**Contribution:** 2
**Rating:** 6
**Confidence:** 4

**Summary:**

This paper introduces a new approach to address the numerical stability issue in diffusion models. The authors propose a new noise schedule and parameterization of preconditioning to eliminate the need for time truncation when dealing with the numerical stability of training and inference. The authors demonstrate that their method eliminates the need for time truncation while maintaining performance on CIFAR10 and ImageNet32x32 datasets. The approach achieves comparable or better results than standard diffusion models without requiring time truncation.

**Strengths:**

1. The method is clear and accessible.
2. The proposed method improves FID and NLL at the same time. This is interesting because previous works suggest that improving likelihood often leads to worse FID.

**Weaknesses:**

1. The contribution needs further clarification:
	1. As shown in B.3 in Karras et al.  [1] and A.2 in Zhang et al. [2], the singularity issue at $t=0$ is fundamentally tied to the use of finite training samples. The target data distribution is a mixture of Dirac measures and its score blows up at training samples. So $\mathcal{J}_{SM}$ is unbounded mathematically. It's inherent and cannot be solved by any parameterization alone.
	2. This paper primarily addresses the singularity issue arising from the parameterization of the neural network. There is a class of parameterization to achieve this and I think the authors should discuss that instead of only focusing on one specific case unless there is a strong reason to do so.
	4. The parameterization proposed in this paper essentially delegates the singularity to the neural network, which eventually leverages the regularization posed by the neural network design. As reported in Table 1, I believe this is also a valid approach but the benefits over the time truncation approach are not convincingly demonstrated both theoretically and empirically.
2. The main experiments in Table 1 omit some relevant baselines such as i-DODE ([3]), soft-truncation ([4]).
3. The manuscript requires several technical clarifications:
	1. Equations (10), (13), (16), and (18) should be explicit about which distribution you are taking expectation over. The current notation uses a single $\mathbb{E}$ for three different expectations.
	2. The definition of D is missing, which first appears in Eq. (10).
	3. $\mathcal{J}_{DSM}$ in Eq. (14) needs an expectation.
	4. Inconsistent notation: the integral over $t$ is written as $\mathbb{E}_t$ in Eq. (12) and integral in Eq. (14).
	5. In Eq. (18), the second expectation should be removed and the equality should be an approximation.

[1] : Karras, Tero, et al. "Elucidating the design space of diffusion-based generative models." _Advances in neural information processing systems_ 35 (2022): 26565-26577.

[2] : Zhang, Pengze, et al. "Tackling the Singularities at the Endpoints of Time Intervals in Diffusion Models." _Proceedings of the IEEE/CVF Conference on Computer Vision and Pattern Recognition_. 2024.

[3] : Zheng, Kaiwen, et al. "Improved techniques for maximum likelihood estimation for diffusion odes." _International Conference on Machine Learning_. PMLR, 2023.

[4] : Kim, Dongjun, et al. "Soft Truncation: A Universal Training Technique of Score-based Diffusion Model for High Precision Score Estimation." _International Conference on Machine Learning_. PMLR, 2022.

**Questions:**

1. Line 149, I'm confused about the claim "these coefficients diverge". $f_t$ and $g_t$ are linear function of $t$ in the VP-SDE, why would they diverge? I think the only coefficient that blows up at $0$ is $g_t^2/\sigma_t$.
2. How does Eq. (18) reduce the variance exactly? Can you provide a formal analysis of the variance reduction properties?
3. How does the design of strata affect the variance reduction?

---

### Official Review · Reviewer_cM18 · 2024-11-04

**Soundness:** 2
**Presentation:** 2
**Contribution:** 1
**Rating:** 3
**Confidence:** 4

**Summary:**

Diffusion models are widely used for high-quality image generation by reversing a process that gradually adds noise to data. However, these models face numerical instability near the end of the time continuum, which often requires heuristic truncation—terminating the process early—to maintain stability during training and sampling. This time truncation disrupts the model's rigor and demands extra tuning. To address this, the proposed FullDiffusion framework introduces a modified noise predictor and a novel SDE solver, removing the need for truncation by ensuring stability in training with maximum likelihood and enabling full-time simulation. Experiments on CIFAR-10 and ImageNet-32 demonstrate improved performance in likelihood and FID, establishing FullDiffusion's effectiveness.

**Strengths:**

1. The overall story flow of the paper feels natural. The motivation for no time truncation, at both side of the boundaries.
2. The paper has written a good background on diffusion models.

**Weaknesses:**

1. The biggest problem of the paper is that both the theoretical and empirical settings, under which the paper is investigated, are out of date. I will detail my arguments below.
2. Essentially, the paper proposes to fix two things about diffusion models: 1. the singularity of score function at $t=0$. 2. $\alpha\neq0$ at time 1. Both problems have been addressed in the field. First, we never want to evaluate the model at $t=0$ anyway, since both ODE and SDE will not modify $x$ if simulated at time $t=0$. The sampling is always done at time where the model can be properly trained. Second, $\alpha\sim0$ is often good enough in practice (the SOTA model EDM [1, 2] uses a rather low terminal noise level). Even if one really wants to have a zero SNR, there are countless works that have already proposed so: [3,4,5,6, ...]. In fact, the proposed formulation is a special case of flow matching, just differing in terms of the interpolation equation.
3. Given the previous point. In order to demonstrate the effectiveness of this particular formulation, and the sampling technique, a more careful and thorough empirical comparison is needed. Currently, the mentioned, closely related baselines are not included in the paper. For example, is the interpolation $\sqrt{1-t^2}$ and $t$ better than $1-t$ and $t$ in [3,4]? Even with the weak baseline, the improvements seem to be marginal, and the results are behind SOTA by quite a bit. I am not asking the authors to beat SOTA, but the pool of baselines needs to be expanded, especially in this case, where the theoretical difference is small.
4. The writing of this paper can be improved. I suggest the author include a short description or intuitive understanding of the equations after derivating them. For example, what modification exactly is added to Equation 18 compared to Equation 16? I assume it is more than just a bigger batch size right?...

In all, I feel the paper lacks in terms of proper comparison with the prior works, both in theoretical analysis and empirical signals, and thus I cannot recommend acceptance at this point.


[1] Karras et al. Elucidating the Design Space of Diffusion-Based Generative Models. NeurIPS 2022.

[2] Karras et al. Analyzing and Improving the Training Dynamics of Diffusion Models. CVPR 2024.

[3] Lipman et al. Flow Matching for Generative Modeling. ICLR 2023.

[4] Liu et al. Flow Straight and Fast: Learning to Generate and Transfer Data with Rectified Flow. ICLR 2023.

[5] Albergo et al. Building Normalizing Flows with Stochastic Interpolants. ICLR 2023.

[6] Girdhar et al. Emu video: Factorizing text-to-video generation by explicit image conditioning. ECCV 2024.

**Questions:**

1. Some typos. For example, though at the beginning of line 193, and better at line 454.
2. I do not understand how the x-prediction and v-prediction suffer from numerical instability whereas the parametrization introduced here does not. Are you referring to the division by $\alpha_t$? If so, if you write out your parametrization in terms of $\epsilon$ and $x_t$, you will also encounter the division by 0. The parametrization does not provide any training signal to the network when $t=1$ right? If this is the reason why the proposed method does not suffer, then the same could be done with x and v-prediction as well. I also suggest the authors look into EDM and flow matching's parametrization.

---

### Meta-Review · Area_Chair_FMC7 · 2024-12-16

**Metareview:**

This paper is concerned with the singularity of training and sampling in diffusion models. The most popular approach to address this singularity is through truncation over the diffusion time interval. This work presents an reparametrization technique to address this singularity. In addition, the paper also provides a variance reduction technique for training and a SDE solver for inference. The main criticism is on the incremental contribution. The reviewers think the problem addressed is not essential and existing methods are sufficient. The empirical evidence presented in this paper is insufficient to show the proposed approach is advantageous over existing ones. Moreover, an indepth comparison with literature is lacking.

**Additional Comments On Reviewer Discussion:**

The authors didn’t respond to reviewers’ comments.

---

### Decision · Program_Chairs · 2025-01-22

Reject